# Real-World Analysis of Survival and Treatment Efficacy in Stage IIIA-N2 Non-Small Cell Lung Cancer

**DOI:** 10.3390/cancers16173058

**Published:** 2024-09-02

**Authors:** Eleni Josephides, Roberta Dunn, Annie-Rose Henry, John Pilling, Karen Harrison-Phipps, Akshay Patel, Shahreen Ahmad, Michael Skwarski, James Spicer, Alexandros Georgiou, Sharmistha Ghosh, Mieke Van Hemelrijck, Eleni Karapanagiotou, Daniel Smith, Andrea Bille

**Affiliations:** 1Guy’s and St Thomas’ NHS Foundation Trust, London SE1 9RT, UK; 2Comprehensive Cancer Centre, King’s College, London SE1 9RT, UK

**Keywords:** non-small cell lung cancer (NSCLC), Stage IIIA-N2, real-world data, real-world evidence, chemoradiation, radiotherapy, surgery, chemotherapy, immunotherapy, survival outcomes, multidisciplinary management

## Abstract

**Simple Summary:**

This study focuses on Stage IIIA-N2 non-small cell lung cancer (NSCLC), which occurs when lung cancer has spread to lymph nodes in the centre of the chest but not to distant parts of the body. This type of cancer is difficult to treat, and survival rates remain low despite advances in therapy. The research is based on real-world data from Guy’s Thoracic Cancer Database and aims to enhance our understanding of patient outcomes and identify important factors that impact survival and disease progression. By examining various treatment strategies, the study aims to offer insights that could help tailor treatment approaches to individual patients. The findings underscore the need for personalised care and the potential benefits of incorporating newer therapeutic options, ultimately leading to better survival outcomes for patients with this challenging form of lung cancer.

**Abstract:**

Background: Stage IIIA-N2 non-small cell lung cancer (NSCLC) poses a significant clinical challenge, with low survival rates despite advances in therapy. The lack of a standardised treatment approach complicates patient management. This study utilises real-world data from Guy’s Thoracic Cancer Database to analyse patient outcomes, identify key predictors of overall survival (OS) and disease-free survival (DFS), and address the limitations of randomised controlled trials. Methods: This observational, single-centre, non-randomised study analysed 142 patients diagnosed with clinical and pathological T1/2 N2 NSCLC who received curative treatment from 2015 to 2021. Patients were categorised into three groups: Group A (30 patients) underwent surgery for clinical N2 disease, Group B (54 patients) had unsuspected N2 disease discovered during surgery, and Group C (58 patients) received radical chemoradiation or radiotherapy alone (CRT/RT) for clinical N2 disease. Data on demographics, treatment types, recurrence, and survival rates were analysed. Results: The median OS for the cohort was 31 months, with 2-year and 5-year OS rates of 60% and 30%, respectively. Group A had a median OS of 32 months, Group B 36 months, and Group C 25 months. The median DFS was 18 months overall, with Group A at 16 months, Group B at 22 months, and Group C at 17 months. Significant predictors of OS included ECOG performance status, lymphovascular invasion, and histology. No significant differences in OS were found between treatment groups (*p* = 0.99). Conclusions: This study highlights the complexity and diversity of Stage IIIA-N2 NSCLC, with no single superior treatment strategy identified. The findings underscore the necessity for personalised treatment approaches and multidisciplinary decision-making. Future research should focus on integrating newer therapeutic modalities and conducting multi-centre trials to refine treatment strategies. Collaboration and ongoing data collection are crucial for improving personalised treatment plans and survival outcomes for Stage IIIA-N2 NSCLC patients.

## 1. Introduction

Non-small cell lung cancer (NSCLC) accounts for approximately 85% of all lung cancer diagnoses [1], with over one-third of patients presenting with locally advanced disease at diagnosis [2]. Stage III NSCLC, a subgroup associated with a poor prognosis [3], is marked by heterogeneity in tumour size, location, and the extent of lymph node metastasis. The 8th edition of the TNM staging system further divides Stage III into IIIA, IIIB, and IIIC [2,4,5], with respective 5-year survival rates of 36%, 26%, and 13% [2]. Stage IIIA-N2 NSCLC, characterised by metastases to ipsilateral mediastinal or subcarinal lymph nodes, lacks distant metastases, permitting consideration of radical treatment. Indeed, the updated 2021 ESMO clinical guidelines [6] have reclassified Stage IIIA under the ‘early stages’ group, previously I-II.

Mediastinal lymph node involvement in NSCLC is complex, with subcategories such as single-station N2, multi-station N2, and unsuspected N2 disease discovered during surgery [7,8]. This heterogeneity complicates treatment decisions, further exacerbated by the lack of clear definitions for resectability, which leads to regional variations in treatment approaches [9,10]. Treatment strategies vary depending on patient characteristics, medical expertise, and available resources, with surgery preferred for resectable cases. For unresectable cases, chemoradiation, often with adjuvant immunotherapy, is typically administered. Recent clinical trials in immunotherapy and targeted therapies have expanded treatment options, though no single regimen has been proven superior [11,12,13,14,15]. These developments highlight the need for personalised, multidisciplinary management approaches.

The absence of standardised protocols and the lack of randomised controlled trials (RCTs) directly comparing treatment strategies pose challenges for clinicians in making evidence-based decisions for Stage IIIA-N2 NSCLC. Real-world data (RWD) offer valuable insights into outcomes across a more diverse and clinically relevant patient population, including those not meeting strict RCT criteria. This study utilises RWD to evaluate overall survival (OS) and progression-free survival (PFS) in Stage IIIA-N2 NSCLC, contributing to the ongoing efforts to refine and personalise treatment strategies for this challenging disease.

### Objectives

The primary objective of this study is to assess the 5-year survival rate, overall survival (OS), and progression-free survival (PFS) in patients with Stage IIIA-N2 NSCLC. Additionally, the study aims to explore the impact of baseline clinicopathological characteristics (e.g., single-station vs. multi-station N2 disease and unsuspected vs. potentially resectable N2 disease) and treatment modalities on long-term outcomes. Notably, the distinction between unsuspected and potentially resectable N2 disease has not been addressed in previous randomised controlled trials (RCTs), highlighting the importance of this study in providing new insights into these subgroups.

## 2. Materials and Methods

### 2.1. Study Design

This study is an observational, single-centre, non-randomised, retrospective analysis of consecutive clinical and pathological T1/2 N2 (TNM 8th edition) non-small cell lung cancer (NSCLC) patients treated with curative intent from 2015 to 2021. Eligible patients were those diagnosed between 1 January 2015 and 1 October 2021, including all patients regardless of their survival duration to address potential survival bias. Patients initially staged with the TNM 7th edition were restaged according to the TNM 8th edition. Clinical N2 disease was diagnosed using PET with or without EBUS.

### 2.2. Patient Cohort

Inclusion criteria comprised a confirmed histological diagnosis of NSCLC, Stage IIIA-N2 classification according to the IASLC 8th edition staging system, receipt of first-line treatment with curative intent, age 18 years or older, and the availability of complete data for analysis. Patients were excluded if they had missing clinicopathological, treatment, or follow-up data.

The study included 142 patients, comprising three groups:Group A—Surgery for clinical N2, 30 patients.Group B—Unsuspected N2 at surgery, 54 patients.Group C—Chemoradiation or radiotherapy alone (CRT/RT) for clinical N2, 58 patients.

### 2.3. Data Collection and Variables

Data were extracted from the retrospective component of Guy’s Thoracic Cancer Real-World Database. Assessed variables included demographics (age, gender), smoking status, Eastern Cooperative Oncology Group (ECOG) performance status, lung function, histology, staging, treatment types, recurrence, and survival rates. Data on surgical approach, type of resection, pathological TNM stage, histopathological data (e.g., lymphovascular and pleural invasion), and resection status were collected for surgical patients. For the radiotherapy cohort, details on dose (gray, Gy), fractions, duration (days), and fractionation schedule were analysed. SACT data included agent (s) used, intent, and duration of treatment.

### 2.4. Outcome Measures 

The primary objective was to evaluate survival outcomes, including overall survival (OS) and progression-free survival (PFS), in patients treated radically for Stage IIIA-N2 NSCLC. The secondary objective was to analyse OS and PFS between the three treatment groups and examine the influence of baseline clinicopathological characteristics (e.g., single-station vs. multi-station N2 disease) on long-term outcomes. Outcome measures included PFS, OS, and recurrence (local, regional, or distant). PFS was measured from the start of treatment to the first date of recurrence or death, and OS from the beginning of treatment to death from any cause. Local recurrence was defined as a lesion at the ipsilateral lung staple line (s) for surgical patients and unequivocal growth at the treated site for radiotherapy patients. Regional recurrence was defined as tumour progression adjacent to the planning target volume, surgical resection margin, or in regional lymph nodes.

### 2.5. Statistical Analysis

Descriptive statistics were employed to report baseline clinicopathologic characteristics. Categorical variables were reported as frequencies and percentages, while continuous variables were represented as mean and range. When the assumptions for the chi-square test were not met, categorical data were compared using Fisher’s exact test. Continuous variables were compared using the Mann–Whitney U test when assumptions for parametric tests were not met. Survival intervals were estimated using the Kaplan–Meier method, with the Log Rank test comparing survival distributions between groups. Results were reported as hazard ratios (HR) with 95% confidence intervals (CI) and *p*-values from the likelihood ratio test. 

### 2.6. Multivariate Cox Regression Models

Univariate and multivariate disease progression/death risk analyses and overall survival were conducted using Cox proportional hazards regression. For the multivariate Cox proportional hazards regression analysis, variables were included based on clinical relevance and previous literature indicating their potential impact on overall survival (OS) in patients with Stage IIIA-N2 NSCLC. A backward elimination procedure was applied to refine the model. This approach involved fitting the full model with all candidate variables and then iteratively removing the least significant variable, one at a time. Following each removal, the model was re-evaluated to ensure that the Bayesian Information Criterion (BIC) did not indicate a deterioration in model fit.

### 2.7. Overall Survival in All Patients

The Cox regression model included the following factors: treatment group, gender, age at diagnosis, smoking status, Eastern Cooperative Oncology Group (ECOG) performance status, lung function (FEV1% predicted and TLCO % predicted), histological subtype, and extent of clinical N2 disease (single or multi-station).

### 2.8. Overall Survival in Surgical Patients (Groups A & B)

The Cox regression model included the following additional factors: procedure type, surgical approach, presence of lymphovascular invasion, presence of visceral pleural invasion, resection status, number of lymph nodes removed, number of positive lymph node stations, and administration of neoadjuvant or adjuvant treatment.

### 2.9. Overall Survival in CRT/RT Patients (Group C)

The Cox regression model included the following additional factors: radiotherapy treatment regimen, radiation dose and fractionation, and platinum agent used.

Statistical significance was set at *p* < 0.05; all *p*-values were two-sided. Stata software (version 18.0) was utilised for all statistical computations and analyses.

## 3. Results

### 3.1. Patient Demographics and Baseline Characteristics

This study involved 142 patients diagnosed with Stage IIIA-N2 NSCLC, categorised into three groups based on their treatment and disease presentation (Figure 1). Group A consisted of 30 patients who had surgery for clinical N2 disease. Group B included 54 patients with unsuspected N2 disease found during surgery. Group C comprised 58 patients treated with chemoradiation or radiotherapy alone (CRT/RT) for clinical N2 disease. The demographic and baseline characteristics of these patients can be found in Table 1.

The mean age, gender distribution, and smoking status showed no significant differences across the groups. However, there was a significant difference in ECOG performance status (*p* < 0.001), with Group C having more patients with poorer performance status (ECOG PS 2 and 3). Lung function measures (FEV1 and TLCO) were similar across groups.

A significant difference was found in histological subtypes (*p* = 0.031). Group B had a higher incidence of adenocarcinoma (83%) compared to Group A (67%) and Group C (55%). Conversely, Group C had a higher incidence of squamous cell carcinoma (36%) compared to Group A (27%) and Group B (15%). The distribution of single-station versus multi-station N2 disease showed a trend towards significance (*p* = 0.099), with Group A having a higher proportion of single-station N2 disease (90%) compared to Group C (72%).

### 3.2. Treatment Characteristics for Group A and Group B (Surgical Patients)

Group A comprised 30 patients who had surgery for clinical N2 disease, while Group B included 54 patients with unsuspected N2 disease found during surgery.

### 3.3. Surgical Approach and Type

In Group A, 23% of patients underwent Robotic-Assisted Thoracoscopic Surgery (RATS), 27% underwent Video-Assisted Thoracoscopic Surgery (VATS), and 50% had open surgery. In Group B, 17% of patients had RATS, 61% had VATS, and 19% had open surgery. Notably, lobectomy was the most common type of surgery in both groups, with 93% in Group A and 83% in Group B. Sublobar resections were performed exclusively in Group B, accounting for 11% of the cases.

### 3.4. Neoadjuvant and Adjuvant Treatments

Table 2 summarises the neoadjuvant and adjuvant treatments patients in Groups A and B received.

Although neoadjuvant chemoimmunotherapy was not licensed for resectable lung cancer during the time period of this study, one patient received this treatment for a concurrent upper gastrointestinal malignancy before surgical resection of both conditions. One patient in Group B received induction stereotactic ablative radiotherapy following a delay to surgery caused by an anaphylactic reaction to anaesthesia.

### 3.5. Pathological Findings

The mean number of lymph node stations removed was five in Group A and four in Group B. Positive lymph node stations averaged three in Group A and 2 in Group B. Lymphovascular invasion occurred in 43% of Group A and 48% of Group B, while visceral pleural invasion was noted in 33% of Group A and 50% of Group B. R0 resection status was achieved in 90% of Group A and 89% of Group B.

### 3.6. Treatment Characteristics for Group C (CRT/RT for Clinical N2 Disease)

Group C comprised 58 patients who underwent chemoradiation or radiotherapy alone for clinical N2 disease. The treatment details of these patients are provided below in Table 3.

Most patients (91%) undergoing concurrent chemoradiation received a dose of 64 Gy in 32 fractions. One patient discontinued treatment due to hospitalisation for a cerebral vascular accident.

Among patients receiving concurrent chemoradiation, cisplatin, and vinorelbine were the most commonly used chemotherapy agents (59%). The most frequently used regimen in the sequential chemoradiation group was carboplatin and pemetrexed (40%).

### 3.7. Survival Analysis 

#### 3.7.1. Disease Free Survival

The cohort was followed for a median period of 26 months (range: 1 to 100 months). During this time, the overall median disease-free survival (DFS) for patients was 18 months (range: 1 to 94 months). At the 2-year mark, 43% of patients remained disease-free, while the 5-year DFS rate dropped to 21%. Group A had a median DFS of 16 months, Group B had 22 months, and Group C had 17 months. These differences were not statistically significant (*p* = 0.99).

The Kaplan–Meier survival curves for DFS among the three groups are presented in Figure 2. The Cox proportional hazards model showed no significant difference in DFS among treatment groups (HR = 1.00, 95% CI: 0.77 to 1.30). The proportional hazards assumption was tested using Schoenfeld residuals, and the global test showed no violation (chi2 = 0.00, *p* = 0.9994).

#### 3.7.2. Overall Survival

The median overall survival (OS) for the entire cohort was 31 months (range: 1 to 100 months), with 2-year and 5-year OS rates of 60% and 30%, respectively. Group A had a median OS of 32 months, Group B had 36 months, and Group C had 25 months. These differences were not statistically significant (*p* = 0.99). Kaplan–Meier survival curves for OS are shown in Figure 3.

The Cox proportional hazards model for OS showed no significant differences between treatment groups (HR = 1.00, 95% CI: 0.76 to 1.31). The proportional hazards assumption held true for this model, with a global test result of chi2 = 2.53 and *p* = 0.1116.

#### 3.7.3. Multivariate Cox Regression Analysis for Overall Survival in All Patients

The multivariate Cox regression analysis aimed to identify factors influencing overall survival (OS) in patients with Stage IIIA-N2 NSCLC, considering baseline and clinical characteristics. Significant variables in the final model included patient age at diagnosis, ECOG performance status, lung function (TLCO % predicted), histology and extent of clinical N2 disease.

Patients with an ECOG performance status of 3 had a significantly higher hazard of death compared to those with an ECOG performance status of 0 (HR = 23.22, 95% CI: 1.98–272.04). Older age at diagnosis was also associated with an increased risk of death (HR = 1.06, 95% CI: 1.01–1.10). Patients with histological subtypes classified as “Other” faced a significantly higher risk of death compared to those with adenocarcinoma (HR = 5.90, 95% CI: 1.52–22.93).

Other factors, including treatment group, gender, smoking status, lung function, and the extent of clinical N2 disease (single-station vs. multi-station), did not show significant associations with survival outcomes.

The disease-free survival (DFS) analysis mirrored the OS results. However, gender was a significant predictor of DFS, with female patients having a higher hazard of disease progression or death compared to male patients (HR = 2.46, 95% CI: 1.08–5.60).

#### 3.7.4. Multivariate Cox Regression Analysis for Overall Survival in Surgical Patients (Group A & B)

A multivariate Cox proportional hazards regression analysis identified significant predictors of overall survival (OS) among surgical patients in Groups A and B. The final model included ECOG performance status at diagnosis, lymphovascular invasion (LVI), and visceral pleural invasion.

Patients with LVI had a significantly higher risk of death (HR = 9.02, 95% CI: 3.15–25.82). Conversely, the presence of visceral pleural invasion was associated with a significantly lower risk of death (HR = 0.37, 95% CI: 0.15–0.91). Patients with an ECOG performance status of 3 had a higher hazard of death compared to those with a status of 0 (HR = 4.37, 95% CI: 1.48–12.88), while those with a status of 2 did not show a statistically significant difference (HR = 1.20, 95% CI: 0.50–2.89).

Other factors, such as patient age, procedure type, surgical approach, resection status, lymph node involvement, and the use of induction or adjuvant treatments, did not demonstrate statistically significant associations with survival outcomes.

To further illustrate the impact of LVI on overall survival, a Kaplan–Meier survival curve was plotted comparing patients with and without LVI (Figure 4).

#### 3.7.5. Multivariate Cox Regression Analysis for Overall Survival in CRT/RT Patients (Group C)

A multivariate Cox proportional hazards regression analysis for Group C (CRT/RT for clinical N2 disease) identified no significant predictors of overall survival after applying the backward elimination procedure.

### 3.8. Recurrence Patterns 

The recurrence patterns were categorised into local, regional, and distant recurrences among the three group. Table 4 shows the distribution of recurrence types in each group.

A chi-square test was used to compare recurrence distributions among the groups. The Pearson chi-square value was calculated as 7.2884 with a *p*-value of 0.121, indicating no significant differences in recurrence patterns. However, there were trends indicating that Group C had the highest local recurrence rate.

## 4. Discussion

Managing Stage IIIA-N2 non-small cell lung cancer (NSCLC) is challenging due to its diverse nature and the need to balance localised and systemic therapies. This study assessed survival outcomes in 142 patients treated with curative intent, categorised into three groups: Group A (surgery for clinical N2 disease, n = 30), Group B (unsuspected N2 disease at surgery, n = 54), and Group C (chemoradiation or radiotherapy alone for clinical N2 disease, n = 58).

The median overall survival (OS) for the entire cohort was 31 months, with 2-year and 5-year OS rates of 60% and 30%, respectively. The median disease-free survival (DFS) was 18 months for the entire cohort. No significant survival advantage was found among the different treatment strategies, emphasising the need for personalised treatment approaches. Significant predictors of overall survival included age, Eastern Cooperative Oncology Group (ECOG) performance status, histological subtype and lymphovascular invasion. At the same time, the extent of clinical N2 disease did not significantly correlate with survival outcomes. 

Our findings align with several studies on the management of Stage IIIA-N2 NSCLC. The EORTC 08,941 study evaluated patients with unresectable Stage IIIA-N2 NSCLC who responded to induction chemotherapy [11] and found no statistically significant difference in median progression-free survival (PFS) or OS between radical radiotherapy and surgical resection groups. Notably, the EORTC study had a high pneumonectomy rate associated with increased postoperative mortality, unlike our study, where pneumonectomy rates were significantly lower (1%).

Similarly, the Intergroup 0139 trial compared preoperative chemoradiation (CRT) followed by surgical resection with definitive CRT in patients with low-volume mediastinal T1–3 N2 NSCLC [12]. This trial reported a statistically significant improvement in PFS for the trimodality treatment arm (12.8 months vs. 10.5 months), but this did not translate into a significant difference in 5-year survival (27% vs. 20%; OR 0.63, 95% CI 0.36–1.10). Similar to the EORTC study [11], high mortality rates were observed among patients undergoing pneumonectomy. 

Meta-analyses of randomised controlled trials have also failed to demonstrate a clear superiority of one treatment regimen over another in resectable Stage III NSCLC [16], consistent with our findings.

Interestingly, unlike the IASLC staging project [17], which showed a statistically significant difference in 5-year survival favouring single-station N2 disease over multi-station N2 disease, our study did not find a significant correlation between the extent of clinical N2 disease and survival outcomes. This discrepancy may be attributed to differences in patient populations or treatment approaches, or the smaller sample size in our study.

Histological subtype significantly impacted survival outcomes in our study. Adenocarcinomas, typically found in peripheral lung regions, were more amenable to surgical resection, leading to higher surgical intervention rates in these cases. In contrast, squamous cell carcinomas, often located centrally, posed higher surgical risks and were more frequently treated with CRT/RT in Group C. This distribution of histological subtypes across the treatment groups suggests an interaction between histology and treatment approach that may influence the survival outcomes observed. Furthermore, our findings indicate that squamous cell carcinomas are less likely to present with occult mediastinal disease compared to adenocarcinomas, as reflected by the higher incidence of adenocarcinoma in Group B.

The finding that visceral pleural invasion was associated with a lower risk of death is counterintuitive, given that pleural invasion is generally considered an adverse prognostic factor [18]. This result may be due to more aggressive surgical intervention in patients with pleural invasion, leading to better outcomes. However, the sample size and study design limit the generalisability of this finding, warranting further investigation with larger cohorts.

The findings of this study have important implications for managing Stage IIIA-N2 NSCLC. The observed outcomes suggest potential benefits in expanding joint surgery and clinical oncology appointments for more technically resectable and operable patients, facilitating multidisciplinary discussions and decision-making.

Advancements in imaging and diagnostic techniques, the increased use of adjuvant immunotherapy after concurrent radiotherapy, and the recent approval of neoadjuvant chemo-immunotherapy before surgery represent significant progress in the treatment of Stage IIIA-N2 NSCLC. However, integrating these advancements into routine practice requires adaptation by multidisciplinary teams.

Previously, only a small proportion of UK patients with Stage III NSCLC received radical intent multi-modality therapy, with preoperative chemotherapy followed by surgical resection being used in <1% of patients [10]. The UK is not accustomed to the routine use of neoadjuvant treatments for NSCLC [10], so integrating neoadjuvant chemo-immunotherapy will require multidisciplinary teams to adapt their practices and become familiar with these new treatment protocols. Ongoing monitoring and comparison of outcomes are essential to determine if these new treatments improve survival rates and overall patient outcomes.

This study has several strengths, including its comprehensive analysis of a large group of Stage IIIA-N2 NSCLC patients treated with curative intent, providing valuable insights into real-world outcomes. However, limitations such as potential biases in retrospective analysis, small sample sizes in subgroups, and the single-centre design should be acknowledged. Additionally, the lack of data on newer treatment modalities like immunotherapy limits the applicability of our findings to the current treatment landscape.

## 5. Conclusions

This study highlights the complex and heterogeneous nature of Stage IIIA-N2 NSCLC, emphasising that no single superior treatment strategy exists among surgery, chemoradiation, and their combinations. It was found that age, performance status, histological subtype and lymphovascular invasion were predictors of overall survival, while the extent of clinical N2 disease did not correlate with survival outcomes. These findings underscore the need for personalised treatment approaches and multidisciplinary decision-making in managing Stage IIIA-N2 NSCLC. Future research should focus on integrating newer therapies and conducting multi-centre trials to refine treatment strategies and improve patient outcomes. Additionally, the search for other predictive markers and the importance of various ‘omics’ fields—such as genomics, proteomics, and radiomics—should be prioritised to better tailor treatments to individual patients and enhance overall survival and quality of life.

## Figures and Tables

**Figure 1 cancers-16-03058-f001:**
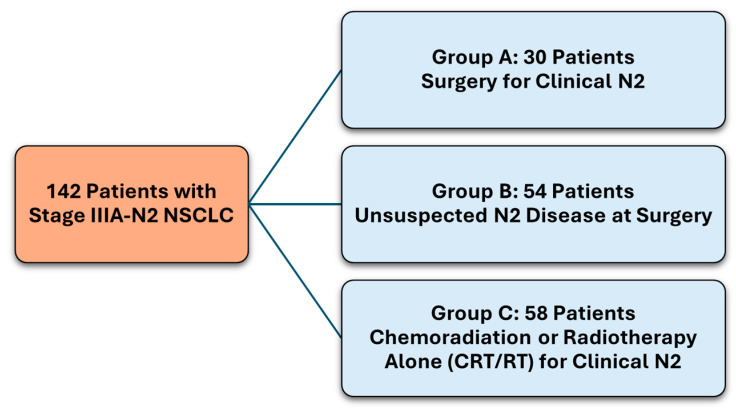
Study cohort distribution.

**Figure 2 cancers-16-03058-f002:**
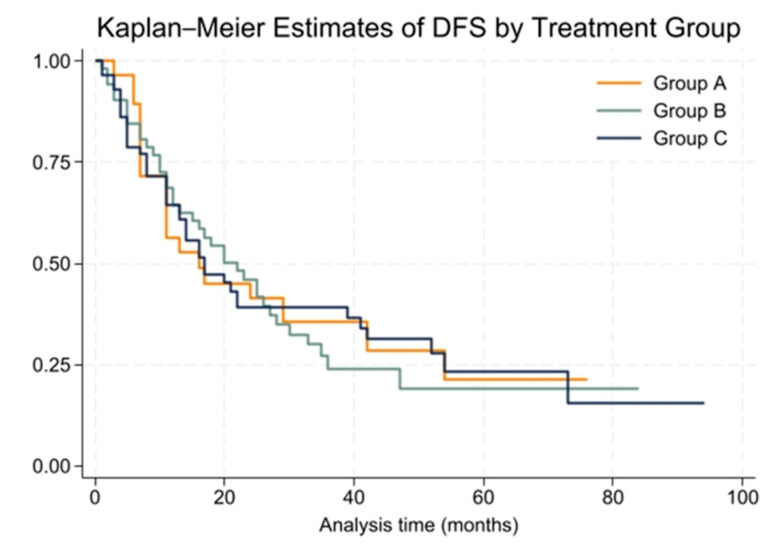
Kaplan–Meier survival curves for disease-free survival (DFS) by treatment group.

**Figure 3 cancers-16-03058-f003:**
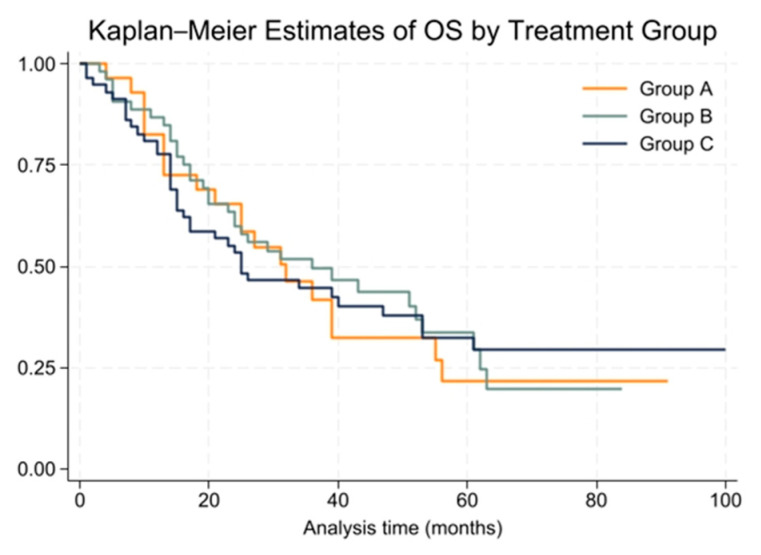
Kaplan–Meier survival curves for overall survival (OS) by treatment group.

**Figure 4 cancers-16-03058-f004:**
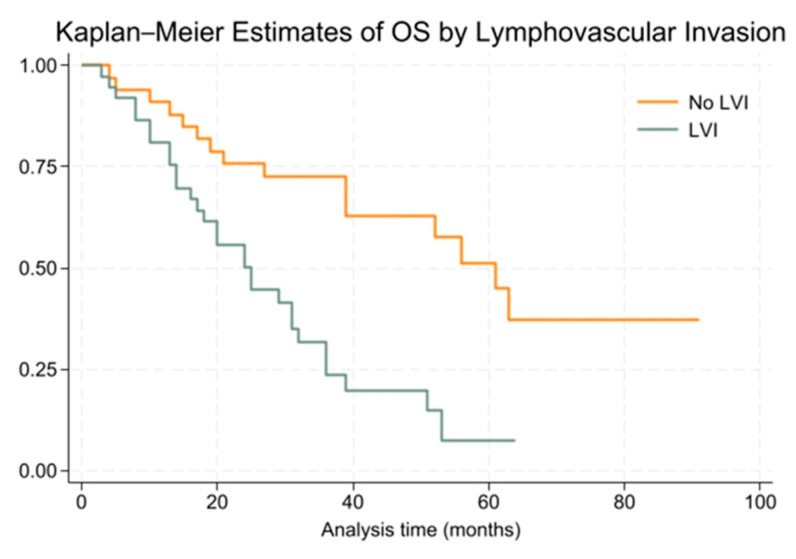
Kaplan–Meier survival curve for overall survival based on lymphovascular invasion (LVI) status for surgical cohort (Groups A and B).

**Table 1 cancers-16-03058-t001:** Demographic and baseline characteristics of patients.

		All Patients	Group A	Group B	Group C	*p*-Value
Age (years), mean		69	68	69	70	0.781
Gender						0.633
	Male	70 (49%)	15 (50%)	24 (44%)	31 (53%)	
	Female	72 (51%)	15 (50%)	30 (56%)	27 (47%)	
Smoking Status						0.293
	Never smoker	14 (10%)	0 (0%)	9 (17%)	5 (9%)	
	Ex-smoker	72 (51%)	17 (57%)	26 (48%)	29 (50%)	
	Current smoker	51 (36%)	11 (37%)	18 (33%)	22 (38%)	
	Unknown	5 (4%)	2 (7%)	1 (2%)	2 (3%)	
ECOG performance status						<0.001
	0	33 (23%)	5 (17%)	16 (30%)	12 (21%)	
	1	55 (39%)	10 (33%)	15 (28%)	30 (52%)	
	2	22 (15%)	4 (13%)	4 (7%)	14 (24%)	
	3	1 (<1%)	0 (0%)	0 (0%)	1 (2%)	
	Unknown	31 (22%)	11 (37%)	19 (35%)	1 (2%)	
FEV1 (% predicted), mean		86	91	86	83	0.324
TLCO (% predicted), mean		73	72	74	73	0.900
Histological subtype						0.031
	Adenocarcinoma	97 (68%)	20 (67%)	45 (83%)	32 (55%)	
	Squamous	37 (26%)	8 (27%)	8 (15%)	21 (36%)	
	NSCLC, NOS	2 (2%)	0 (0%)	0 (0%)	2 (3%)	
	Other	6 (4%)	2 (7%)	1 (2%)	3 (5%)	
Clinical N2 Disease						0.099
	Single-station	69 (78%)	27 (90%)	NA	42 (72%)	
	Multi-station	19 (22%)	3 (10%)	NA	16 (28%)	

ECOG: Eastern Cooperative Oncology Group, FEV1: Forced Expiratory Volume in 1 s, TLCO: Transfer Factor of the Lung for Carbon Monoxide.

**Table 2 cancers-16-03058-t002:** Neoadjuvant and adjuvant treatment.

	Group A (n = 30)	Group B (n = 54)
Neoadjuvant treatment	7 (23%)	1 (2%)
Neoadjuvant chemotherapy	2	0
Neoadjuvant chemo-immunotherapy	1	0
Neoadjuvant chemo-radiotherapy	4	0
Neoadjuvant radiotherapy	0	1
Adjuvant treatment	15 (28%)	40 (74%)
Adjuvant chemotherapy	9	22
Adjuvant targeted therapy	0	1
Adjuvant chemo-radiotherapy	4	17
Adjuvant radiotherapy	2	0

**Table 3 cancers-16-03058-t003:** Treatment characteristics for Group C.

Treatment Modality	Number of Patients (%)
Concurrent chemo-radiotherapy	26 (45%)
Concurrent chemoradiotherapy + adjuvant immunotherapy (Durvalumab)	6 (10%)
Sequential chemo-radiotherapy	15 (26%)
Radiotherapy alone	11 (19%)

**Table 4 cancers-16-03058-t004:** Distribution of recurrence types by group.

Type of Recurrence	Group A (n = 30)	Group B (n = 54)	Group C (n = 58)
Local	1 (3%)	0 (0%)	5 (9%)
Regional	7 (23%)	14 (26%)	18 (31%)
Distant	10 (33%)	24 (44%)	16 (28%)
Total	18 (59%)	38 (70%)	39 (68%)

## Data Availability

The datasets presented in this article are not readily available to protect confidential information. Requests to access the datasets should be directed to the corresponding author.

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
