# Peer review of "Real-World Analysis of Survival and Treatment Efficacy in Stage IIIA-N2 Non-Small Cell Lung Cancer"

_cancers, 2024, doi:10.3390/cancers16173058_

Round 1

Reviewer 1 Report

Comments and Suggestions for Authors

Dear authors, we express our gratitude for the opportunity to be reviewed by you. I believe that your research is interested in the absence of randomized controlled trials. The length of the introduction section is excessive, and the inclusion of subsections is ineffective. Kindly eliminate them and enhance the introduction's uniqueness and smoothness. Kindly remove the Chinese text and correct the phrase "(Chemo)radiotherapy." Also, please review the discussion that is too lengthy and repetitive.

Author Response

Comments 1:
Dear authors, we express our gratitude for the opportunity to be reviewed by you. I believe that your research is interested in the absence of randomized controlled trials. The length of the introduction section is excessive, and the inclusion of subsections is ineffective. Kindly eliminate them and enhance the introduction's uniqueness and smoothness. Kindly remove the Chinese text and correct the phrase "(Chemo)radiotherapy." Also, please review the discussion that is too lengthy and repetitive.

Response 1: 

Dear Reviewer, Thank you for your thoughtful and constructive feedback on our manuscript. We have carefully considered your comments and made the following revisions:
  1. Introduction Section:
    • Reduction of Length and Removal of Subsections: The introduction has been revised to eliminate the subsections, as suggested. The revised introduction is now more concise and we have streamlined the content to improve the flow of the introduction while ensuring that the essential background information is retained. 
    • Correction of Terminology: The phrase "(Chemo)radiotherapy" has been corrected to "chemoradiation or radiotherapy alone (CRT/RT)" to accurately reflect that some patients did not receive the chemotherapy component.
  2. Removal of Chinese Text:
    • The unintended appearance of Chinese characters in the text has been corrected. We have reviewed the entire manuscript to ensure that no other such errors are present.
  3. Discussion Section:
    • Reduction of Length and Elimination of Repetition: We acknowledge that the discussion section was overly lengthy and, at times, repetitive. We have streamlined the discussion to focus on the key findings and their implications while reducing redundancy. 
We hope these revisions address all your concerns and enhance the quality of the manuscript. We appreciate your time and effort in reviewing our work.

Reviewer 2 Report

Comments and Suggestions for Authors

Congratulations to you on your novel study of the comparison of the treatment of Stage IIIA-N2 lung cancer. The manuscript was well written. I think this article will be a good reference of the future clinical practice. It is good to be considered for publication after correction of some typos, e.g. in line 53, subdivides Stage III disease into stage IIIA, IIIB and III ; also in line 322 appearance of Chines characters

Author Response

Comments 1: Congratulations to you on your novel study of the comparison of the treatment of Stage IIIA-N2 lung cancer. The manuscript was well written. I think this article will be a good reference of the future clinical practice. It is good to be considered for publication after correction of some typos, e.g. in line 53, subdivides Stage III disease into stage IIIA, IIIB and III ; also in line 322 appearance of Chines characters

Response 1: Thank you very much for your kind words and positive feedback regarding our manuscript titled “Real-World Analysis of Survival and Treatment Efficacy in Stage IIIA-N2 Non-Small Cell Lung Cancer.” We are pleased that you found our study to be a valuable contribution to the field and potentially beneficial for future clinical practice.

We have carefully reviewed your comments and have made the necessary corrections as per your suggestions:

  1. Correction of Typos:
    • We have corrected the typo in line 53, where "Stage III disease" is now correctly subdivided into "Stage IIIA, IIIB, and IIIC" instead of "III".
    • The appearance of Chinese characters in line 322 has been removed, and the text has been reviewed to ensure no other unintended characters or errors are present.

We believe these revisions have further improved the clarity and accuracy of our manuscript. We sincerely appreciate your constructive feedback and the time you have taken to review our work.

Reviewer 3 Report

Comments and Suggestions for Authors

The manuscript by Josephides et al., dealing with the Real-World Analysis of Survival and Treatment Efficacy in Stage IIIA-N2 Non-Small Cell Lung Cancer. This study focuses on Stage IIIA-N2 non-small cell lung cancer (NSCLC), which occurs when lung cancer has spread to lymph nodes in the center of the chest, but not to the distant parts of the body. This study indicates that this type of cancer is difficult to treat. Also, the survival rates remain low despite the advances of the available therapy. The data this study obtained from Guy’s Thoracic Cancer Database with aim to improve the survival rate among NSCLC. The findings underscore the need for personalized care and the potential benefits of incorporating newer therapeutic options, ultimately leading to better survival outcomes for patients. The manuscript is well written, and the data are represented in best for. Accordingly, the manuscript can be published in the present form

Author Response

Comments 1: The manuscript by Josephides et al., dealing with the Real-World Analysis of Survival and Treatment Efficacy in Stage IIIA-N2 Non-Small Cell Lung Cancer. This study focuses on Stage IIIA-N2 non-small cell lung cancer (NSCLC), which occurs when lung cancer has spread to lymph nodes in the center of the chest, but not to the distant parts of the body. This study indicates that this type of cancer is difficult to treat. Also, the survival rates remain low despite the advances of the available therapy. The data this study obtained from Guy’s Thoracic Cancer Database with aim to improve the survival rate among NSCLC. The findings underscore the need for personalized care and the potential benefits of incorporating newer therapeutic options, ultimately leading to better survival outcomes for patients. The manuscript is well written, and the data are represented in best for. Accordingly, the manuscript can be published in the present form

Response 1: Thank you for your thorough review and positive feedback on our manuscript titled “Real-World Analysis of Survival and Treatment Efficacy in Stage IIIA-N2 Non-Small Cell Lung Cancer.” We are delighted that you found our study well-written and that the data are represented effectively.

Your endorsement of the manuscript in its present form is encouraging, and we are grateful for your support. Thank you once again for your valuable time and insights.